# LONG TAIL CLASSIFICATION THROUGH COST-SENSITIVE LOSS FUNCTIONS

## ABSTRACT

Class imbalance in the data introduces significant challenges in training machine learning models especially with long-tailed datasets, i.e., where a small number of classes comprise a large number of sample data points in the training data. Specifically, it leads to biased models that overfit with respect to the dominant classes while underfitting on the minority classes. This, in turn, results in seemingly satisfactory yet biased overall results. Hence, the above biasing needs to be controlled such that the desired generalizability of the model is not compromised. To that end, we introduce a novel Cost-Sensitive Loss (CSL) function designed to dynamically adjust class weights, and incorporate a reinforcement learning mechanism to optimize these adjustments. The proposed CSL function can be seamlessly integrated with existing loss functions, to enhance their performance on imbalanced datasets, rendering them robust and scalable. We implemented the above CSL function in form of a framework which leverages reinforcement learning to optimally apply these adjustments over consecutive training epochs. Experimental results on benchmark datasets demonstrate that our proposed approach significantly outperforms state-of-the-art methods. The results indicate that our approach can provide an optimal trade-off between model accuracy and generalization while training models on imbalanced data.

## 1 INTRODUCTION

In the ever advancing field of Machine Learning, availability of balanced datasets with equal representation of all classes is a requirement for creating a reliable model usable in real-life cases. While this scenario looks quite natural, the reality couldn't be more different. During data collection in real-life scenarios, it is almost impossible to curate a perfect dataset. There occurs a motley of imbalances Yang et al. (2022) in the representations of classes in the dataset. Class imbalance in the data introduces significant challenges in training machine learning models especially with long-tailed datasets Yang et al. (2022). Specifically, it leads to biased models that overfit with respect to the dominant classes while underfitting on the minority classes. This, in turn, results in seemingly satisfactory yet biased overall results. Hence, the above biasing needs to be controlled such that the desired generalizability of the model is not compromised.

Long-tailed datasets are characterized by a few dominant (majority) classes compared to a overwhelmingly vast number of under-represented classes (minority), leading to models that exhibit skewed performance. Models trained on such datasets excel in predicting the majority classes while often failing to adequately generalize to minority classes. This issue is not merely academic but holds profound implications for practical applications where accurate and unbiased predictions across all classes are essential, such as medical diagnostics, fraud detection, and anomaly detection in industrial processes.

Several research efforts have addressed this issue Zhang et al. (2023). Approaches for dealing with long-tail classification can be broadly categorized into information augmentation techniques Kim et al. (2020); Park et al. (2022); Wang et al. (2021a); Han et al. (2005); Zhong et al. (2021a), cost-sensitive learning (CSL) Lin et al. (2017); Cui et al. (2019); Ren et al. (2020); Cao et al. (2019); Park et al. (2021); Legate et al. (2023) and model improvement methods Dong et al. (2017); Huang et al. (2016); Liu et al. (2019a); Zhu & Yang (2020); Ouyang et al. (2016); Zhong et al. (2021b); Zhou et al. (2020); Wang et al. (2020); Zhang et al. (2022). Data-level techniques (also known as

information augmentation (IA) techniques) rely on changing the underlying data distribution by either oversampling the minority classes or undersampling the majority classes to reduce the bias and improve generalization. Yet these approaches introduce either new types of biases or substantially increase computational costs. Therefore they can hardly be applied to large, real-world datasets. Model improvement (MI) approaches, such as meta-learning, focus more towards generalization of models from imbalanced data, which often necessitates huge computational resources and a substantial amount of labeled data, a requirement that, in most real-world situations, is quite infeasible.

Amidst these approaches, Cost-Sensitive Learning (CSL), which is also known to as Class-Sensitive Learning in some literature, emerges as a promising paradigm. CSL alters the standard loss function to penalize prediction errors differently depending on class distribution. This would allow the model to enhance its generalization and prediction performance without actually changing the underlying data. CSL is very attractive because it attacks the problem at the model level where, indeed, it can nudge learning representations differently. CSLs still depend on static weight schemes, and thus are not adaptive to the dynamic nature of real-world datasets. During training as the complexity of learning for each class changes, the static weights would not catch such nuances, making them less efficient.

To that end, we introduce a novel CSL function designed to adapt dynamically to the distribution of samples within each class. The above function incorporates a mechanism to adjust class weights based on the effective proportion of classes in given samples. Our framework leverages reinforcement learning to optimally apply these adjustments over consecutive training epochs. Through a deeper understanding of the insights regarding the interplay between class representation and model prediction accuracy derived from empirical results, our framework aims to enhance both the robustness and generalization capabilities of machine learning models on imbalanced datasets. The contributions of this paper are summarized as follows.

1. We introduce a new CSL function which dynamically re-weights the samples in accordance with features learned of the classes along with learning complexity of the classes, therefore leads to improvements in generalization to minority classes.

2. We implement this CSL function within a framework that easily integrates it into different machine learning architectures, thereby improving their performance on imbalanced datasets, making them more scalable and robust.

3. We demonstrate the efficacy of our CSL function through extensive experimentation on benchmark long-tailed datasets like CIFAR-10-LT, CIFAR-100 LT, and ImageNet-LT, improving model accuracy and generalization over state-of-the-art methods. The code and the datasets for this paper are available at icl.

## 2 METHOD

Cost-Sensitive Learning methods are easier to implement with no extra modules being added to the model and incur a very slight increase in training time. MI and IA methods show promising and state-of-the-art results but can lead to a significant increase in the training time. Our framework retains the simplicity of existing Cost-Sensitive Learning techniques while significantly improving the performance of the model. We propose the design of a novel Cost-Sensitive Learning (CSL) function that allows us to dynamically adapt class weights based on features learned and the complexity of learning them for a particular class. The complexity of learning features for a particular class is measured in terms of the entropy gain from that class in a particular epoch.

Our approach also leverages reinforcement learning to apply these weight-adjustments over successive training epochs. It is done by calculating the loss at each epoch and *rewarding the model* with a reward value 'k' depending on the performance improvement it made compared with the previous epoch. Our CSL function is a novel re-weighting approach that surpasses existing CSL functions, in terms of performances achieved on benchmark datasets under similar training conditions. Our CSL function is inspired by L2 regularization. It minimizes the loss while paying equal "importance" to learning the tail classes despite less samples. We use a factor $\gamma_i$ for each class $i$ that signifies the importance (or rather unimportance; higher $\gamma_i$, less importance is given to class $i$) assigned to that class. This factor is multiplied with $N_{\text{pred},i}$, the number of times class $i$ was predicted in the current epoch after validation, as explained below. The intuition is that, if a class $i$ has been pre-

dicted more (less), we penalize it by diminishing (increasing) its importance in terms of reducing the term $\gamma_i \times N_{\text{pred},i}$. Thus, if a dominant class is predicted more, because its features have been better represented by the model due to the larger number of training samples, its importance is then reduced so that the model can focus on learning tail classes subsequently. Our method penalizes the model for misclassification of samples depending on the weights assigned to each class that signify their importance.

Our CSL function considers (1) How well a class was learned in the previous epoch as well the complexity of learning the class in terms of the entropy gain from that class, (2) The number of times a particular class was accurately predicted during validation in this epoch (signifying how well the model has represented the features of that class), and (3) An adjustable reinforcement term which is decided by comparing the performance of the model in the $i^{th}$ epoch with that in the $(i-1)^{th}$ epoch. The last term is inspired by Reinforcement Learning policies of learning from previous experiences, adjusting the parameters of current epoch based on the experience of the model during training in the previous epoch. The CSL function is expressed in terms of the following hyperparameters.

$N_{pred,i}$: denotes the total number of times the class $i$ was predicted by the model during its validation in this epoch. Our method aims at empowering the model to increase the $N_{\text{pred},i}$ value for the tail classes. This also increases the risk of overfitting, which is controlled using techniques explained below.

$\gamma_i$: This value determines the importance assigned to class $i$ and is expressed in terms of the semantic values and the complexity of (learning the features of) the class. It is used to direct the model-training in a way that forces the value of $N_{\text{pred},i}$ to increase for the tail classes while keeping the value almost consistent for other classes, based on how well the model has learned the features of the classes. This is a dynamic parameter which is updated after each epoch based on the accuracy observed while learning the features of each class in the previous epoch.

*Reinforcement term*: We add a constant reward-term $reinforcement\_term$, which quantifies an additional increment to the loss function depending on the level of improvement in the training in the current epoch compared to the previous epoch. This term tunes the value of the loss function to prevent the model to converge without getting stuck at a local optimum.

Our CSL function can be added as an additional term to any loss function. In our evaluation, we use it alongside Cross Entropy loss given as

$$\mathcal{L}(y_{\text{true}}, y_{\text{pred}}) = -\sum_{i=1}^{C} y_{\text{true},i} \log y_{\text{pred},i} + \frac{1}{C} \sum_{i=1}^{C} \frac{(\gamma_i \cdot N_{\text{pred},i} + \text{reinforcement\_term})^2}{\sum_k (z_k - e_i)^2 + \epsilon} \quad (1)$$

where $C$ is the number of classes, $z_k$ is the predicted value for the $k$th data point, $e_i$ is the true value associated with the $i$th class, and $\epsilon$ is a small constant added to prevent division by zero and ensure numerical stability. During training, as the model encounters more datapoints from the dominating classes, the "credit" for the decrease in the loss function goes more to the dominating classes. Thus the features of the dominating classes get better representation than those of the tail classes. The "semantic values" (i.e., learned feature storage) Ma et al. (2023) of the dominating classes are thus larger than those of the tail classes. The importance $\gamma_i$ for the $i$th class is computed based on these semantic values Ma et al. (2023) as well as the complexity of learning features from that class (measured in terms of the increase in entropy). The $\gamma_i$ values computed for the dominating classes are thus higher than those for the tail classes. This forces the model to reduce the number of predictions for the dominating classes while increasing the number of predictions of the tail classes in the subsequent epochs to ensure that the term $\gamma_i \cdot N_{\text{pred},i}$ reduces. In turn, this results in better learning of the features of the tail classes subsequently (i.e., the model focuses more on learning the features of the tail classes). Of course, the strategy of of increasing the $N_{\text{pred},i}$ values of tail classes, as the training proceeds, risks overfitting to the tail classes. This is prevented by multiplying $N_{\text{pred},i}$ with $\gamma_i$. Using the semantic value Ma et al. (2023) as a metric to calculate $\gamma_i$ for each class ensures that the model will not increase the $N_{\text{pred},i}$ value of the tailed classes just as a way of reducing the loss. As the training proceeds, the number of learned features of tail classes will increase, and so will the gamma values. This would ensure that the increase in $N_{\text{pred},i}$ values are kept in check and that datapoints are not predicted from the tail classes solely to lower the loss value. This ensures that the model keeps increasing the $N_{\text{pred},i}$ value of tailed classes only if it wishes to learn the tail classes better, and is not driven by the sole motivation to lower the loss value.

The $\gamma_i$ values have a dynamic nature, changing their values every epoch according to how many features of that particular class $i$ were learned in the previous epoch.

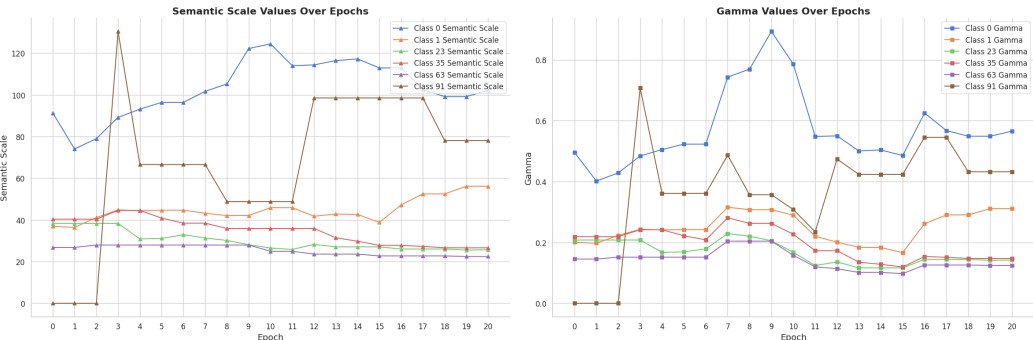

Figure 1: Semantic Scale Values          Figure 2: Gamma Values

Figures 1 and 2 are plots of semantic values depicting the feature storage of each class $i$ and $\gamma_i$ values assigned to each class for the first 80 epochs during the training of ResNet32 on CIFAR 100 with stochastic gradient descent (SGD) (momentum = 0.9) and a imbalance ratio [1] of 100. Following the training strategy adopted by state-of-the-art Tan et al. (2020) for Long Tail Learning via cost sensitive loss, the initial learning rate was set to 0.1 and then decayed by 0.01 at 160 epochs and again at 180 epochs. We visualize the importance of the dynamic nature of the weights ($\gamma_i$) assigned to each class. As proposed, the $\gamma_i$ values change depending on the changes in the learned feature storage, that is, how well a class is learned. Existing methods consider the number of samples of each class to penalize the model. This strategy assigns higher weights to the tail classes which are maintained constant throughout the training process Cui et al. (2019).

Our CSL function assigns weights $\gamma_i$ to the classes considering how difficult it is to learn their features. In Figures 1 and 2 we can see the model learns class 91, bicycle in CIFAR 100, much better irrespective of lesser number of samples in the class, due to which a higher $\gamma_i$ is assigned to it which forces the model to learn features of other difficult to learn classes which have been assigned lower $\gamma_i$. Similarly, for epochs 8 to 11, the feature storage for class 91 is constant. That is when $N_{\text{pred},i}$ and reinforcement term comes into picture, increasing the loss which lowers the $\gamma_i$ value of class bicycle till the 11th epoch, pushing the model to prioritize learning its features until we see a rise in the feature storage graph on the 12th epoch. This strategy is followed for all the classes while the training proceeds. It adds a dynamic nature to the training procedure prioritizing the classes on the basis of the difficulty to learn them.

## 3 ALGORITHM

Our CSL function works with dynamic parameters and deals with imbalanced datasets, learning from how well the classes are being learned, and how well the model is performing in its previous epochs of training. Our CSL function (Equation-2) is given by (the CSL function added to the cross entropy loss function was shown in Equation 1):

$$\frac{1}{C} \sum_{i=1}^{C} \frac{(\gamma_i \cdot N_{\text{pred},i} + \text{reinforcement\_term})^2}{\sum_k (z_k - e_i)^2 + \epsilon} \tag{2}$$

where parameters $C$, $z_k$, $e_i$, and $\epsilon$ are as defined in Equation 1. During training, consider the scenario of calculating the total loss for the model in $n$th epoch (for illustration purposes, we assume that the CSL function has been added to a cross-entropy loss function). First, the probabilistic interpretation is obtained by calculating the cross-entropy loss function, measuring the differences between the true labels' probability distribution and the predicted probability distribution output by the model. The cross-entropy loss is denoted by $L(y, f(x,w)) = -\sum_{k=1}^{C} y_k \log f_k$, where $y_k$ is a ground truth,

---

[1] ratio of the number of samples in the (largest) majority class to the number of samples in the (smallest) minority class

and $f_k$ is the k-th output of the model $f(x, w)$ with parameters $w$, with $C$ being the total number of classes. Algorithm 1 shows how our CSL function works when added to a loss function (in this case cross-entropy). After the cross-entropy loss is computed, we compute the feature storage for each class $i$.

---

1: **Algorithm 1**
2: **Input**: training dataset $D = (X, Y)$, number of classes $C$, target class index $T$
3: **Output**: model trained with custom loss, additional term, and entropy-based gamma
4:
5: **Phase 1: Feature collection and initialization**
6: Initialize the model with random parameters $w$
7: Initialize feature storage for all classes $\mathcal{F}_i$ where $i = 1$ to $C$
8: Initialize gamma values $\gamma$, class prediction history $\mathcal{H}_p$, semantic scales history $\mathcal{H}_s$
9: Initialize entropy storage $\mathcal{E}_i$ for all classes
10:
11: **Phase 2: Forward pass and loss calculation**
12: **for** each epoch $t = 1$ to $T$ **do**
13:    **for** each mini-batch $(x, y) \in D_m$ **do**
14:       1. Predict class probabilities $p = f(x, w)$
15:       2. Compute cross-entropy loss $L_{CE} \leftarrow -\sum_y y \log(p)$
16:       3. Store features in $\mathcal{F}_i$ for all classes
17:       4. Compute semantic scales $S_i$ for all classes
18:       5. Calculate entropies $H_i$ for all classes based on their predictions
19:       6. Calculate gamma values $\gamma_i \leftarrow \frac{S_i}{(1e-6+\max(S)\cdot H_i)}$ for all classes
20:       7. Store gamma values, class predictions, and entropy history
21:       8. Compute additional term $T_{add} \leftarrow \sum_i \frac{(\gamma_i \cdot N_{C_i} + \text{reinforcement})^2}{\sum (\text{inputs} - \text{one-hot encoded class vector})^2 + \epsilon}$
22:       9. Normalize the additional term $T_{add} \leftarrow \tilde{T}_{add}/C$
23:       10. Update loss $L_{total} \leftarrow L_{CE} + T_{add}$
24:    **end for**
25: **end for**

---

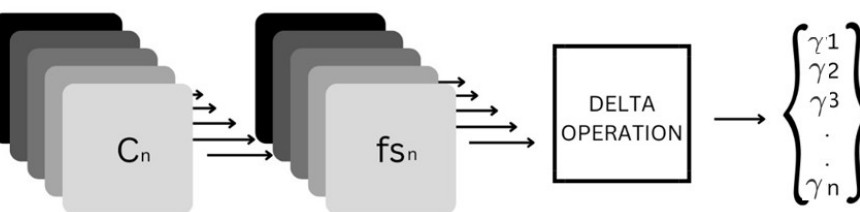

Figure 3: Explaining the procedure of each part of the CSL function where $C_n$ in the diagram represents class $n$, $f_{S\_n}$ in the diagram represents the learned feature-storage of $C\_n$ class when passed from Delta operation provides us with the allotted $\gamma_n$ for that class

More specifically, the algorithm maintains a record of class-specific features and adjusts the loss based on these recorded features. The feature volume, or semantic scale, for each class is then calculated from the feature storage by counting the number of stored predictions. More precisely, the semantic scale $S_i$ is computed as the square of the average magnitude of the feature vectors $f_{ij}$, represented as $S_i = \left(\frac{1}{N_i} \sum_{j=1}^{N_i} \|f_{ij}\|\right)^2$ Ma et al. (2023). This count is used to determine the dynamic gamma values, which scale the importance of each class relative to the class with the maximum feature volume.

These semantic values are passed via a DELTA function (see Figure 3) which combines the semantic values learned from each class with the complexity of learning the features of that class, represented with the entropy gained from the class (see below). Thus $\gamma_i$ is computed as $\gamma_i = \frac{S_i}{(1+\epsilon)(H_i \cdot \max(S_i))}$ where $\gamma_i$ are the weights for each class and $H_i$ is the entropy gained from class $i$. Lastly an additional term $\sum_k (z_k - e_i)^2$ is added to the denominator of the CSL function to make it differentiable while back-propagation. Finally, the CSL function is normalized (by dividing it by the number of classes) before adding it to the cross-entropy loss calculated earlier.

The complexities of each class are computed as entropies $H_i$ for each class based on their predictions in Algorithm 1. We have empirically studied the reason to consider the complexity of each class before directly assigning the weights on the basis of feature storage. Existing research Cui et al. (2019); Park et al. (2021) have correlated the number of samples in each class with the weights assigned to them. When feature storage is correlated with the weights to be assigned, we must take the complexity to learn a particular class into consideration. An Easier To Learn (ETL) class is recognised with its data points closely distributed with less variance, for example, *Airplane* in CIFAR 10 (there are only few types of Airplanes). However, a class with sparsely distributed points with higher variance is Difficult To Learn (DTL) such as *Dog* in CIFAR 10 (there is an enormous variety in breeds and types of a natural object like dog). The model can learn an ETL class quickly despite it belonging to the tail part of the distribution whereas sometimes having more samples of DTL class might not be enough for the model to learn its features. To increase the correlation of feature storage with weights, we need to avoid a situation where an ETL class belonging to the tail part is not assigned a lower weight, and ensure that DTL classes are assigned lower weights, so that the model can focus on learning the features of the DTL classes while the training procedure continues.

The entropy $H_i$ for each class $i$ serves as an effective measure of the complexity of learning features of that class. For ETL classes, the model provides confident predictions and hence a provides a lower value of entropy function. For DTL classes, the model is more uncertain with datapoints spread out resulting in a higher value of the entropy. The Delta function is inversely proportional to the complexity since we require lower $\gamma_i$ values for classes $i$ with higher complexity. After we have received the weights $\gamma_i$ for each class $i$, the loss function uses $\gamma_i$ along with $N_{\text{pred},i}$ and the reinforcement_term to compute the CLS function.

## 4 IMPLEMENTATION

For the implementation of our CSL function, we have used the PyTorch framework and selected different model architectures suited for each dataset. For fair and accurate comparisons with previous works, we follow the same settings. For CIFAR-10 and CIFAR-100, we used ResNet-32 with random weight initialization as our backbone. We trained the model for 200 epochs using Stochastic Gradient Descent (SGD) as the optimizer, with a momentum of 0.9, an initial learning rate of 0.1, and a weight decay of $2 \times 10^{-4}$ to prevent overfitting. We also used a learning rate scheduler to reduce the learning rate at the 160th and 180th epochs for better model convergence.

For ImageNet LT dataset we used ResNet-50 model, trained with SGD optimiser with 0.9 momentum and 5e-4 weight decay with an initial learning rate of 0.01 and batch size 256 along with a cosine scheduler. Long tailed version of ImageNet is constructed using Pareto distribution Liu et al. (2019b). We also evaluated our method on Tiny ImageNet to compare with some previous works. For Tiny ImageNet we used ResNet-18 as our backbone. Input images are resized to 224 × 224 and loaded with a batch size of 128. SGD optimizer is used whose learning rate is initially set to 0.05 and decays using a cosine scheduler with the weight decay and momentum being 5e-4 and 0.9, respectively.

## 5 EVALUATION

To effectively evaluate our proposed CSL loss function, we ran experiments using four commonly used benchmark datasets: CIFAR-10, CIFAR-100, ImageNet-LT Liu et al. (2019b) and Tiny ImageNet at imbalance ratio 100. CIFAR-10 and CIFAR-100 are considered standard balanced datasets for image classification, consisting of 60,000 32 × 32 color images of 10 and 100 classes, respectively. We introduced imbalance to these datasets by varying the number of samples for each class. The number of selected samples in the $k$-th class was set to $n_k \mu^k$, where $n_k$ is the original number of samples in the $k$-th class and $\mu$ is a parameter in the range $(0, 1)$ with the imbalance ratio $(p)$ ranging from 10 to 200. The imbalance ratio is calculated as: $p = \frac{\max_k(n_k)}{\min_k(n_k)}$. where $k$ varies from 1 through the number of classes. Tiny ImageNet was used which contains 100K images from 200 categories, with class strength of 500 samples each. The same mechanism as described above was used to induce imbalance in this dataset, for imbalance ratio 100. ImageNet LT is a long tailed dataset, a subset of ImageNet dataset with 115.8K images from 1000 categories. This dataset has a maximum

of 1280 images per class and a minimum of 5 images per class. It reflects the real world where class distributions are highly skewed, making it an ideal benchmark for CSL methods on long-tailed datasets.

## 5.1 BASELINES

To effectively understand how well our proposed CSL loss function performs, we compared it against several established and well-known methods, including Cross-Entropy Loss (CE) Zhang & Sabuncu (2018), Focal Loss Lin et al. (2018), Class-Balanced Loss (CB Loss) Cui et al. (2019), LDAM (Large Margin Softmax Loss) Cao et al. (2019) and Influence Balanced loss Park et al. (2021). Each method has shown effectiveness in different aspects, making them good benchmarks for evaluating our approach. CE Loss is the most commonly used method for classification and helps us clearly understand how much our method has improved compared to traditionally used methods. Focal Loss is designed to give importance to low-sample classes, focusing on high imbalance. These baseline methods were chosen because they are well-established in the field and each offers a different approach to solving the long-tailed imbalance problem, which helps us clearly assess our performance. We also compare our results with well other known methods from the field of Module Improvement (MI) methods to compare how well our CSL function can bridge the gap between the performances recorded by CSL functions and that offered by advanced techniques like OLTR Liu et al. (2019b) (MI), Decoupled-CB-CRT Kang et al. (2020) (MI) and LFME Xiang et al. (2020) (MI).

## 5.2 RESULTS

| Method | Classes in CIFAR-10 | | | | | | | | | | |
|---|---|---|---|---|---|---|---|---|---|---|---|
| | Plane | Car | Bird | Cat | Deer | Dog | Frog | Horse | Ship | Truck | Avg_acc |
| #Training Samples | 5000 | 3237 | 2096 | 1357 | 878 | 568 | 368 | 238 | 154 | 100 | |
| **Baseline (CE)** | 97.4 | 98.0 | 84.0 | 80.3 | 78.8 | 68.4 | 76.1 | 64.5 | 57.0 | 52.0 | 74.8 |
| Focal | 91.6 | 95.1 | 73.1 | 59.2 | 67.2 | 84.2 | 77.3 | 74.3 | **83.9** | 61.8 | 76.8 |
| CB | 92.9 | 96.3 | 79.2 | 75.1 | 72.7 | 69.5 | 70.6 | 75.3 | 73.3 | 66.8 | 78.1 |
| LDAM | **96.9** | **98.3** | 74.7 | 72.1 | 82.4 | 69.9 | 75.0 | 73.0 | 64.3 | 66.0 | 76.8 |
| LDAM-DRW | 94.8 | 97.8 | **82.6** | 72.3 | **85.3** | 73.0 | 82.0 | 76.7 | 75.8 | 72.4 | 80.9 |
| IB | 92.2 | 96.2 | 81.3 | 66.6 | 85.7 | 76.4 | 81.7 | 75.9 | 79.9 | **81.1** | 81.70 |
| IB + CB | 93.8 | 97.2 | 78.1 | 64.8 | 84.4 | 76.2 | **86.4** | **79.7** | 79.5 | 76.9 | 81.54 |
| IB + Focal | 90.9 | 96.1 | 81.7 | 69.0 | 82.0 | 75.7 | 85.2 | 77.5 | 80.2 | 76.8 | 81.51 |
| **CSL_Ours** | 96.26 | 93.75 | 79.33 | **87.64** | 85.27 | **78.4** | 79.49 | 74.39 | 78.77 | 69.89 | **82.31%** |

**Table 1:** Class-wise performance comparison on CIFAR-10 dataset at imbalance ratio 50 in ResNet-32 for different methods, including Influence Balanced loss Park et al. (2021), for comparison of performance.

| | CIFAR 10 | | CIFAR 100 | |
|---|---|---|---|---|
| | $p = 100$ | $p = 50$ | $p = 200$ | $p = 100$ |
| Methods | Avg accuracy | Avg accuracy | Avg accuracy | Avg accuracy |
| CE | 70.4 | 74.8 | 34.84 | 38.32 |
| CE + CB | 72.4 | 78.1 | 26.23 | 38.6 |
| Focal + CB | 74.6 | 79.3 | 35.62 | 39.6 |
| LDAM-DRW | 77 | 80.9 | - | 42 |
| Focal | 70.4 | 76.7 | - | 38.4 |
| LDAM | 73.4 | 76.8 | - | 42 |
| LDAM-DRW + SSP | 77.83 | 82.13 | - | 43.43 |
| **CSL Ours** | **78%** | **82.31%** | **49.13%** | **52.01%** |

**Table 2:** Comparison of different methods on CIFAR-10 and CIFAR-100 with various imbalance ratios using Top 1 accuracy; (-) represents no experiments are available from their previous study. $p$ represents imbalance ratio

To evaluate the performance, extensive experiments were conducted on all four datasets, with different levels of class imbalance and the performance was compared with existing methods based

| Methods | Type | ImageNet Accuracy |
|---|---|---|
| CE | CSL | 41.6 |
| Weighted Softmax | CSL | 49.1 |
| ESQL | CSL | 48 |
| Focal loss | CSL | 47.2 |
| OLTR | MI | 46.7 |
| Decoupled-CB-CRT | MI | 44.9 |
| LFME | MI | 47 |
| **CSL Ours** | CSL | **49.3** |

**Table 3:** Comparison of models performance, trained on recognized cost-sensitive learning based loss functions on ImageNet LT for 200 epochs.

| Methods | $p = 100$ |
|---|---|
| Baseline (CE) | 38.52 |
| Focal | 38.95 |
| LDAM | 37.47 |
| **CSL Ours** | **39.47%** |

**Table 4:** The above results are of Tiny ImageNet trained on ResNet-18 adopted from Park et al. (2021) paper. $p$ represents imbalance ratio

on re-weighting cost sensitive learning. We experimented with imbalance ratios of 50 and 100 on CIFAR-10, 100 and 200 on CIFAR-100, on ImageNet LT (ImageNet LT is long-tailed version of ImageNet constructed using Pareto distribution Liu et al. (2019b)), and on Tiny ImageNet with imbalance ratio 100. Previous studies had compared class-wise accuracy of the model on CIFAR-10 at 50 imbalance ratio, trained on different baseline methods following re-weighting strategy. Table 1 shows that our CSL function provides the best accuracy over the baselines along with significantly higher accuracies on DTL classes such as Cat. To analyze the performance of ResNet 32 with different levels of imbalance, we created long-tailed versions of CIFAR 10 and CIFAR 100 using the formula $n_k \mu^k$ with different values of $\mu$. Table 2 documents results of studies with different levels of imabalance for CIFAR 10 and CIFAR 100. We observe our CSL function performs better than the state-of-the-art. This improvement shows the effectiveness of our CSL function in tuning the parameters of the loss function according to the distribution of the features learned by the model during training. Based on the penalty assigned to the classes on the basis of the class distribution in previous epoch, the CSL function successfully shifts the focus of the model from frequently encountered classes to tail and difficult to learn classes.

Tables 3 and 4 document our results on ImageNet-LT and Tiny ImageNet at imbalance ratio of 100 Here, we compare our CSL function-based approach with other existing studies that use CSL functions Liu et al. (2019b); Xiang et al. (2020); Kang et al. (2020), including the state-of-the-art Weighted softmax loss function Wang et al. (2021b). We also compare with some MI techniques in Table 3. One can see in Tables 3 and 4 that our CSL function outperforms the state-of-the art, both for ImageNet LT and Tiny ImageNet.

# 6 RELATED WORK

**Long Tailed Recognition and Imbalance Learning:** Long tailed learning is a common problem faced in machine learning where the training dataset contains a disproportionately large number of data points for a very small fraction of the classes. Many real-world datasets typically follow a long-tailed distribution, which makes the model biased towards the dominant or *head* classes, while producing relatively degraded performance on the minority or *tail* classes. Therefore, deep learning models trained using traditional approaches are unable to produce good results in real-world applications. Existing research has mostly focused on increasing the accuracy of the model on tail-classes based on the number of samples in each class Yang et al. (2022); Zhang et al. (2023). We focus on adaptively training the model while taking into account the proportion of different classes in the training data. According to Zhang et al. (2023), existing body research in this area can be divided into the following categories according to the methodology they follow.

**Information Augmentation:** Information augmentation (IA) methods are based on generating new (or duplicate) samples for the tail classes to compensate for the disproportionate number of head classes. These may use GANs or transformers to synthesise tail samples of the tail class to increase the models' accuracy over the tail part of datasets, while introducing more uncertainty in the resultant sample deficient classes. One such method named Major-to-Minor translation (M2m), presented in Kim et al. (2020), is a head-to-tail transfer augmentation technique which employs an

approach similar to that of adversarial attacks to tweak the features comprising the samples of the head class such that they resemble samples of the tail class. In Park et al. (2022), the authors propose to make the minority samples more diverse through a simple technique of image pasting. In Wang et al. (2021a), the authors introduce a framework named RSG (Rare-class sample generator) that performs dynamic augmentation of the features of the tail class. It updates the features of the tail class with the difference in values of the features in the data points comprising the head class with respect to the center of that class. The above difference incorporates variance in the features that helps create a diverse enough dataset that is capable of sufficient generalization on the tail classes infrequently occurring in the sample data points. SMOTE Han et al. (2005) represents another class of IA approaches, referred to as non-transfer augmentation, which leverages oversampling techniques, such as MiSLAS Zhong et al. (2021a). It observed that mixing up of sample data has positive and negative effect on representation learning and classifier learning, respectively, and minimize the chances of model over-confidence.

**Module Improvement:** Module Improvement (MI) methods are based on improving deep neural network architectures in long-tailed learning. They can be categorized into representation learning, classifier design, decoupled training, and ensemble learning.

**Representation Learning:**

- Metric Learning: This technique is based on distance metrics specific to certain categories of tasks for feature spaces that are discriminative in nature. Prior works such as Large Margin Local Embedding (LMLE) Huang et al. (2016) learns by determining intra-cluster and inter-class margins. The work on Class Rectification Loss (CRL) Dong et al. (2017) mines hard pairs for adjusting weights of tail-classes.

- Prototype Learning: This approach is based on learning feature definitions specific to each class. The Open Long-Tailed Recognition (OLTR) Liu et al. (2019a) framework augments features based on visual meta memory. Likewise, the Inflated Episodic Memory (IEM) Zhu & Yang (2020) framework performs dynamic update on the memory blocks to improve feature definitions.

- Sequential Training: This methodology is based on sequential training, i.e., it continually learns the features from the representations of the data samples that it encounters during successive training epoch. Among existing techniques in this category, the Hierarchical Feature Learning (HFL) Ouyang et al. (2016) uses hierarchical clusters for feature transfer. Another method, namely Unequal-training Zhong et al. (2019), separates the head-class and tail-class portion from the training dataset, and performs training separately on these.

**Classifier Design:** The most commonly used classifier in deep learning is $\phi(w^T \cdot f + b)$, which suffers from the problems of long tailed distribution. Recent studies have proposed ways to deal with the imbalance at classifier level. Along with linear and cosine classifier, complicated classifier methods like RTC Wu et al. (2020) have been proposed to deal with class imbalance by means of hierarchical classification.

**Decoupled Training:** This represents a category of techniques, such as MiSLAS Zhong et al. (2021b), typically train feature extractors for learning representations which can be generalized, and re-train the classifiers to handle class imbalance.

**Ensemble Learning:** This category of techniques combine multiple network architectures to address the problems with long-tailed learning, such as BBN Zhou et al. (2020) which is used for dynamic training. RIDE Wang et al. (2020) follows a unique approach by training concurrent agents (experts) with the full sample data, but maintains variance in the model by means of KL-divergence loss. SADE Zhang et al. (2022), trains different agents for each class distributions, and aggregates the models learned by the agents in an adaptive manner using self-supervised learning.

### 6.0.1 COST-SENSITIVE LEARNING

Cost-Sensitive Learning (CSL) methods augment the training loss, making it more customized for each class with the goal of re-balancing the uneven training caused by imbalance datasets. These methods either re-weight the training loss values for individual classes by multiplying a constant $k$ calculated by their *re-weighting* scheme, or they adjust the loss function by subtracting different margin factors from different classes calculated according to their *re-margining* scheme.

**Re-margining methods**: To handle class-imbalance, these methods reduce the losses by certain margins determined by the corresponding classes, maintaining a minimal margin between features and the respective classifier. Label-Distribution-Aware Margin (LDAM) Cao et al. (2019) sets a different margin for each class based on frequency of the training label, and thus produces a considerably larger margin for the tail classes.

**Re-weighting methods:** These techniques apply easy to use loss functions as follows. *Focal loss* Lin et al. (2018) adds a modulating factor to the cross-entropy that reduces the loss for well-classified examples and increases the loss for the misclassified examples. *Weighted softmax* Wang et al. (2021b) implicates the simplest relation of multiplying the loss values of individual classes with the inverse of their number of samples. *Class-balances loss* Cui et al. (2019) quantifies the relation between the class frequencies and loss for each class, by defining an *effective number* and assuming that the loss for each class is inversely proportional to the effective number of samples. *Balanced Softmax* Ren et al. (2020) proposes adjusting prediction logits by multiplying them by label frequencies. This adjustment helps alleviate class imbalance bias by incorporating the labels prior before computing the final losses.

Module improvement methods are mostly based on class re-balancing techniques, and are complementary to information augmentation methods, typically leading to better performance in real-world applications. However, these methods require variance in the model and incurs larger overhead while training due to additional complex elements (terms) added to model. Cost-Sensitive Learning methods deal with data imbalance in a faster and easy to implement way by developing customized loss functions. This paper is based on Cost-Sensitive Learning that introduces a novel loss function based on penalizing the model on the basis of complexity (entropy) of learning samples from a particular class. In Du & Wu (2023), the authors propose to minimize a loss function that is the geometric mean of the losses of the individual classes instead of averaging them. This loss function is not a CSL function.

## 7 CONCLUSIONS AND FUTURE WORK

This paper provides a scalable and adaptable cost-sensitive learning approach to train accurate models from class imbalanced data while not compromising on generalizability. We validate effectiveness of our framework through extensive experiments on benchmark long-tailed datasets, demonstrating significant performance improvements over state-of-the-art techniques in the area. Our approach dynamically tunes class weights in a novel loss function based on the effective proportion of classes observed in each epoch. We leverage the exploratory mode of operation in reinforcement learning to avoid being stuck in a local minimum while tuning the above class weights. In our future work, we aim to further optimize this CSL function by reducing frequent changes in loss function parameters, to avoid the computation of erratic gradients.

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
