# OpenReview forum: "Long Tail Classification  Through Cost Sensitive Loss Functions"
_ICLR.cc/2025/Conference — ICLR 2025 Conference Withdrawn Submission_

### Official Review · Reviewer_2uej · 2024-10-30

**Soundness:** 2
**Presentation:** 2
**Contribution:** 2
**Rating:** 3
**Confidence:** 4

**Summary:**

This paper introduces a novel Cost-Sensitive Loss (CSL) function to address class imbalance in long-tailed datasets. The CSL function dynamically adjusts class weights and incorporates a reinforcement learning mechanism to optimize these adjustments, significantly improving generalization for minority classes. Experimental results show that the approach outperforms state-of-the-art methods on benchmark imbalanced datasets, achieving a better balance between model accuracy and generalization.

**Strengths:**

1.	The proposed method dynamically adjusts weights based on class complexity, effectively handling class imbalance.
2.	The proposed method uses reinforcement learning to progressively optimize the model.
3.	The proposed method outperforms existing methods on benchmark datasets with significant improvement..

**Weaknesses:**

1. There are numerous formatting issues throughout the paper.
2. The proposed method is empirical and lacks theoretical foundations to ensure its effectiveness.
3. The compared baselines are outdated, all being from before 2020, and lack more recent baselines such as [1-5].

[1]Disentangling label distribution for long-tailed visual recognition. 2021.

[2]Parametric contrastive learning. 2021.

[3]Delving into Semantic Scale Imbalance. 2023.

[4]Long- Tailed Recognition via Weight Balancing. 2022.

[5]Label-Imbalanced and Group-Sensitive Classification under Overparameterization. 2021.

4. From line 374, where it states “(-) represents no experiments are available from their previous study,” it seems that the baseline results may have been directly copied from other papers. A fair comparison should be conducted under the same settings and conditions.
5. The paper lacks experiments on widely used Places-LT and iNaturalist-LT.
6. In addition to the main results, there are no ablation studies or analyses of the proposed method.

**Questions:**

Refer to the weakness part.

---

### Official Review · Reviewer_Nr7d · 2024-10-31

**Soundness:** 2
**Presentation:** 1
**Contribution:** 2
**Rating:** 3
**Confidence:** 3

**Summary:**

This paper focuses on the problem of classification for imbalanced data. It proposes a cost-sensitive learning approach to alleviate the bias caused by learning using only the cross entropy loss.
Specifically, it proposes a regularization term consists of terms representing the importance for each class, and a method to adjust these terms after each epoch during training.
The proposed method is thoroughly examined using standard benchmark datasets and the implementation is public on github.

**Strengths:**

- The paper focuses a problem with a high practical importance in real-world machine learning applications, and also enjoys theoretical interests. The paper succinctly summarizes the existing three approaches to the problem and representative existing methods of each approach.
- The paper pays attention on the regularization approach which enjoys computational efficiency, as discussed in the paper.
- The paper considers an adaptive approach to adjust the class importance according to the learning mechanism.
- Authors share the code implementation which is useful for further investigation.

**Weaknesses:**

- Some writings may draw misunderstanding.
  - For example, "This, in turn, results in seemingly satisfactory yet biased overall results" may not hold if a proper evaluation metric is selected to consider long tail classes.
  - Is "CSLs still depend on static weight schemes" true that none of existing methods ever considered dynamically adjust weights?
  - The word "novel" is used at an overwhelming degrees, even in the title of the github README.
- Although it is an overall easy to follow manuscript, the whole structure is not well organized, causing significant difficulties on understanding the paper.
  - Detailed definitions appear after the first usage of notations.
  - Some concepts are used without clear definition, such as "semantic values" (or "semantic") and "entropy storage".
- Some details are ommited in the summary. For example, the proposed method not only use a regularization term, but also "maintains a record of class-specific features", which is not mentioned in abstract and introduction.
- Training time is a point of CSL, but not sufficiently discussed in the experiment section.

**Questions:**

- "Thus, if a dominant class is predicted more, because its features have been better represented by the model due to the larger number of training samples"
  - A dominant class is predicted more can be caused because the number of data of the dominant class is larger than that of a minor class. Is the the number of data of each class properly considered? $N_{pred, i}$ is not depend on the number of data of class $i$?
- For equation 1, what is the definition of $e_i$? Is the loop over $k$ is calculated for each single data point?

---

### Official Review · Reviewer_J8NP · 2024-11-01

**Soundness:** 2
**Presentation:** 2
**Contribution:** 2
**Rating:** 3
**Confidence:** 4

**Summary:**

The paper proposed a cost-sensitive loss to dynamically reweight each class using reinforcement learning techniques. The proposed CSL function can be integrated with the existing loss function to offer further performance improvements in learning with Long-tailed datasets.

**Strengths:**

1. The problem studied in this paper is valuable.
2. The paper is well-written.
3. The proposed method can be integrated with other loss functions.

**Weaknesses:**

1. The proposed method is complex and challenging to follow. The calculation of the reinforcement term is not explicit.
2. Experiment compared methods are outdated, and state-of-the-art long-tailed learning methods [1,2,3] should be considered.
3. No theoretical analyses are conducted concerning, i.e., convergence.
4. The experimental section is slightly weak. The evaluation in this paper only reported the top-1 accuracy. The accuracy of Many/Medium/Few-shot categories [4-6] is widely used in long-tailed learning. I think comparing different methods on these criteria can further demonstrate the performance of the proposed method, especially the promotion of the tail classes.
5. No sensitivity analysis of hyperparameters was conducted.
6. The results are not reported with an error bar.


[1] Wang, Yidong, et al. "Exploring vision-language models for imbalanced learning." International Journal of Computer Vision 132.1 (2024): 224-237.

[2] Xu, Zhengzhuo, et al. "Learning imbalanced data with vision transformers." Proceedings of the IEEE/CVF conference on computer vision and pattern recognition. 2023.

[3] Ma, Yanbiao, et al. "Curvature-balanced feature manifold learning for long-tailed classification." Proceedings of the IEEE/CVF conference on computer vision and pattern recognition. 2023.

[4] CUDA: Curriculum of Data Augmentation for Long-tailed Recognition. ICLR 2023. https://openreview.net/pdf?id=RgUPdudkWlN

[5] Class-Conditional Sharpness-Aware Minimization for Deep Long-Tailed Recognition. CVPR 2023: 3499-3509. https://openaccess.thecvf.com/content/CVPR2023/papers/Zhou_Class-Conditional_Sharpness-Aware_Minimization_for_Deep_Long-Tailed_Recognition_CVPR_2023_paper.pdf

[6] Distilling Virtual Examples for Long-tailed Recognition. ICCV 2021: 235-244. https://openaccess.thecvf.com/content/ICCV2021/papers/He_Distilling_Virtual_Examples_for_Long-Tailed_Recognition_ICCV_2021_paper.pdf

**Questions:**

1. How is the validation dataset produced during training? Is the validation dataset also long-tailed and evaluated while comparing the model's performance in the $i_{th}$ epoch with that in the $(i−1)_{th}$ epoch?
2. As you mentioned, the term $\gamma_i$ is higher for the majority classes, and as a consequence, the $N_{pred,i}$ will be reduced to reduce their product in the loss function. How could you guarantee that the forced reduced  $N_{pred,i}$ for majority classes will not underfit the majority classes? Have you observed any side effects?
3. How is the entropy $H_i$ calculated during training?
4. To what extent is the proposed CSL approach more efficient than existing IA and MI approaches? Is the proposed method as efficient as other CSL approaches?
5. Could you explain more about calculating the reinforcement term, as you did not show it in the algorithms?
6. How many independent experiments have you repeated to get the numbers shown in your tables? Have you checked the results are statistically significant?

---

### Official Review · Reviewer_7r81 · 2024-11-03

**Soundness:** 3
**Presentation:** 2
**Contribution:** 2
**Rating:** 3
**Confidence:** 4

**Summary:**

This paper tackles the issues of class imbalance in long-tailed datasets, where traditional cost-sensitive learning functions typically rely on static weight schemes that cannot adapt to the dynamic nature of real-world data. To address this limitation, the authors propose a novel CSL function, which adjusts class weights dynamically, leveraging reinforcement learning for optimal weight adjustments over epochs. Extensive experimentation on CIFAR-10, CIFAR-100, and ImageNet-LT datasets reveals that this approach outperforms serveral methods, validating its effectiveness.

**Strengths:**

- Addressing the issue of long-tailed learning is meaningful and highly relevant to real-world applications.
- The idea of dynamically adjusting the class weights is innovative and valuable.

**Weaknesses:**

- The paper is hard to follow, and the writing could be improved. For example, the definition of $e_i$ in Sec. 2 as "the true value associated with the $i$-th class" is confusing.
- The effect of *Reinforcement term* is somewhat unclear and requires further explaination.
- The iNaturalist dataset, a widely adopted large-scale long-tailed dataset, should be included in experiments to further validate the method's effectiveness.
- In the abstract, the authors state that the method "outperforms state-of-the art methods", which appears to be an overstatement. In fact, the baseline is relatively weak. More powerful baselines such as PaCo [1], BCL [2] should be included. Additionally, for cost-sensitive learning functions, the logit adjustment loss [3] is a classic approach and should be added as a baseline.
- The paper lacks an ablation study on the effectiveness of the hyperparameters.
- There are also some typos, such as in line 495, where "Class-balances loss" should be "Class-balanced loss".

-----

1. Parametric Contrastive Learning, ICCV 2021
2. Balanced Contrastive Learning for Long-Tailed Visual Recognition, CVPR 2022
3. Long-tail learning via logit adjustment, ICLR 2021

**Questions:**

Please refer to the weakness section.

---

### Note · Authors · 2026-07-13

**Comment:**

The work has been revised and accepted for publication at a different venue.

**Withdrawal Confirmation:**

I have read and agree with the venue's withdrawal policy on behalf of myself and my co-authors.

---

### Meta-Review · Area_Chair_vzko · 2024-12-17

**Metareview:**

This paper presents a novel Cost-Sensitive Loss (CSL) function that dynamically adjusts class weights with a reinforcement learning mechanism for optimization. The overall comments from our reviewers are negative. The authors didn't take part in the rebuttal phase. I have to reject this paper accordingly. Nonetheless, I hope the reviewers' comments will help improve the quality of this paper.

**Additional Comments On Reviewer Discussion:**

The authors didn't take part in the rebuttal phase.

---

### Decision · Program_Chairs · 2025-01-22

Reject